# Novel Strategy for Non-Aqueous Bioconjugation of Substituted Phenyl-1,2,4-triazole-3,5-dione Analogues

**DOI:** 10.3390/molecules27196667

**Published:** 2022-10-07

**Authors:** Hugh G. Hiscocks, Alison T. Ung, Giancarlo Pascali

**Affiliations:** 1School of Mathematical and Physical Sciences, Faculty of Science, University of Technology Sydney, Broadway, NSW 2007, Australia; 2School of Chemistry, University of New South Wales, Kensington, NSW 2052, Australia; 3Prince of Wales Hospital, Nuclear Medicine and PET, Randwick, NSW 2031, Australia; 4National Imaging Facility, University of New South Wales, Kensington, NSW 2052, Australia

**Keywords:** bioconjugation, sulfur pentafluoride, phenyl-1,2,4-triazoline-3,5-diones

## Abstract

A novel 4-[4-(pentafluoro-λ⁶-sulfanyl)phenyl]-1,2,4-triazole-3,5-dione (**5a**) was synthesised as a potential [^18^F]radio-prosthetic group for radiolabelling peptides and proteins via selective bioconjugation with the phenolic side chains of tyrosine residues. Preliminary conjugation tests revealed the rapid hydrolysis of **5a** under semi-aqueous conditions; these results led to further investigation into the electronic substituent effects of PTAD derivatives and corresponding hydrolytic stabilities. Five derivatives of **5a** with *para* substituents of varying electron donating and withdrawing effects were synthesised for the investigation. The bioconjugation of these derivatives with model tyrosine was monitored in both aqueous and organic media in the presence of a variety of catalysts. From these investigations, we have found HFIP to be an effective catalyst when used in tandem with DCM as a solvent to give PTAD-tyrosine conjugate products (**6a**–**f**) in satisfactory to good yields (54–79%), whereas analogous reactions performed in acetonitrile were unsuccessful. The discovery of this system has allowed for the successful conjugation of electron-deficient PTAD derivatives to tyrosine, which would otherwise be unachievable under aqueous reaction conditions. The inclusion of these electron-deficient, fluorinated PTAD derivatives for use in the PTAD-tyrosine conjugation will hopefully broaden their applicability within fields such as ^19^F-MRI and PET imaging.

## 1. Introduction

Bio-conjugation is the coupling of a biologically active molecule, such as a peptide, protein or antibody, with a functional molecule of a particular function such as a drug, nanoparticle, fluorescent tag or radiolabel. Conjugation between the two components is achieved via an intermediary covalent linkage established by a reactive and chemoselective prosthetic group (Figure 1).

Phenyl-1,2,4-triazoline-3,5-diones (PTADs) represent a promising avenue for the chemo-selective bioconjugation of tyrosine residues. Several aspects concerning PTAD-based bioconjugation are attractive. The PTAD moieties’ chemoselectivity for tyrosine residues, its tolerance to a wide pH range, the rate with which bioconjugation is achieved, and the subsequent conjugate stability. These impressive features have resulted in the growing application of PTAD derivatives as tools for bioconjugation [1,2,3,4,5,6,7].

PTAD derivatives offer excellent chemoselectivity for tyrosine residues; this is evident from the investigations performed by Ban et al., who introduced PTAD to an equimolar solution of C- and N-protected amino acids, bearing potentially reactive side chains, including Cys, Lys, Ser, His, Trp, Arg and Tyr [2]. The authors found that only trace amounts of lysine (Lys) and tryptophan (Trp) were modified, obtaining the desired tyrosine conjugate in 55–58% yields.

However, it should be noted that the chemo-selectivity of PTAD derivatives also depends on their hydrolytic stability, as the corresponding aryl isocyanate is an intermediary hydrolysed byproduct of PTAD derivatives. This intermediate presents a major issue in the context of bioconjugation, as it readily undergoes nucleophilic attack by the terminal amine of lysine residues to form the corresponding urea, resulting in off-target conjugation and potential disruption to biomolecular activity.

Ban et al. first demonstrated the wide pH range within which bioconjugation between tyrosine residues of bovine serum albumin and PTAD can be achieved across a pH range of 2–10 [1]. The authors achieved a yield of 52% at pH = 2, and yields ranging from 85–98% between pH = 7–10. Therefore, the PTAD moiety offers substantial advantages over more commonly utilised pH-sensitive prosthetic groups, such as those that target the terminal thiol or amine of cysteine and lysine side chains. It should be noted that these yields were obtained with electron-rich phenoxy ether PTAD derivatives.

Bioconjugation between PTAD derivatives and Tyrosine side chains can be achieved in under 5 min, a reaction time comparable to CuAAC, SPAAC and tetrazine ligation-type click reactions. This conjugation rate is highly advantageous when time is limited, such as in the context of labelling biomolecules with short-lived radioisotopes, whereas radiosynthesis and bio-conjugation must be performed rapidly to compensate for radioactive decay. The chemical stability of PTAD conjugates was highlighted by Ban et al. in 2010, studying the conjugate of phenyl-1,2,4-triazoline-3,5-dione with the phenolic tyrosine model, p-cresol [2]. When the conjugate was subjected to acidic conditions (10% HCl in MeOH, 24 h), 89% of the conjugate was recovered. When the PTAD-cresol conjugate was subjected to basic conditions (10% NaOH in MeOH, 24 h) and high temperatures (120 °C, neat, 1 h), the conjugate was recovered in the quantitative yield. The impressive stability of the PTAD-tyrosine conjugate is substantially relative to alternative bioconjugate approaches, such as those performed between maleimide derivatives and cysteine, the conjugate of which is both reversible and susceptible to hydrolytic cleavage under basic conditions [1].

Our group has previously demonstrated that the 3- and 4-nitrophenyl sulphur pentafluoride can undergo ^18^F/^19^F radioisotopic exchange. [8] For this reason, our initial investigation of the PTAD moiety for selective bioconjugation with tyrosine residues was centralised around the use of 4-[4-(pentafluoro-λ⁶-sulfanyl)phenyl]-1,2,4-triazolidine-3,5-dione (**5a**) for use as a potential [^18^F]radio-prosthetic group for radiolabelling peptides and proteins.

The advantageous qualities exhibited by the PTAD moiety attracted our interest towards the use of 4-[4-(pentafluoro-λ⁶-sulfanyl)phenyl]-1,2,4-triazole-3,5-dione *(***5a**) as a radioprosthetic group. PTAD derivatives (**5a**–**f**) were prepared from the corresponding anilines (Figure 2). Anilines were first reacted with an excess of carbodiimidazole to form the intermediate N-acyl imidazoles, which were then treated with an excess of hydrazine diethyl carboxylate to form the corresponding N-({phenyl carbamoyl}amino)ethoxyformamide derivatives (**3a**–**f**). Base catalysed cyclisation of these intermediates resulted in the corresponding cyclic urazole derivatives (**4a**–**f**); these were then oxidised to the corresponding PTAD derivatives (**5a**–**f**) using an Oxone, NaNO_2_, SiO_2_ system (Figure 2). Spectral data, synthetic procedures and yields for compounds **3a**–**f**, **4a**–**f** and **5a**–**f** can be found in the Appendix A.

To the author’s knowledge, there have been only two reports of using the PTAD approach to radiolabel biomolecules with Fluorine-18. The first application of a [^18^F]PTAD derivative as a radioprosthetic group for bioconjugation was reported in 2013 by Flagothier et al. [3]. The authors utilised [^18^F]4-[4-(fluoro)]-1,2,4-triazole-3,5-dione as a radioprosthetic group for conjugation with *N*-acyl-L-tyrosine methylamide in semi-aqueous phosphate buffer/acetonitrile solvent, obtaining a yield of 65 ± 5%. Later in 2015, Al-Momani et al. investigated the use of substantially more electron-poor [^18^F]4-[4-(sulfonyl fluoride)]-1,2,4-triazole-3,5-dione, for conjugation with L-tyrosine; however, the authors obtained a yield of only 4.6% under similar semi-aqueous reaction conditions.

The apparent variability in yield of radiofluorinated conjugates obtained between both groups, despite their use of similar reaction parameters, would indicate that the electronic effects of functional groups substituted on the aromatic ring are primarily responsible for the substantial variation in yields. In line with this observation, preliminary conjugation tests revealed the poor performance of **5a** under semi-aqueous conditions; these results prompted us to investigate further the effect of substituent groups on PTAD derivative stability to provide researchers with a better understanding of the PTAD moieties stability and subsequent utility.

## 2. Results

Conjugation between **5a** and model amino acid, *N*-acetyl-*L*-tyrosine (**7**) was initially tested in phosphate-buffered saline (pH = 7.4). After adding **5a** to the reaction mixture (1 equiv), the solution transformed from a fluorescent pink to a light green colour within seconds, indicating consumption of the conjugated PTAD precursor. Analysis of the reaction mixture after 30 min by ^1^H-NMR indicated a product conversion of approximately 5%. No detectable trace of the PTAD precursor was observed, indicating it had been competitively consumed by hydrolysis.

To circumvent the issue of hydrolysis, we then attempted the conjugation under anhydrous conditions; however, when the aqueous component of the solvent system was omitted entirely, no product formation was observed, even after one hour of reaction. These results are consistent with the results obtained by Ban et al., who found that the reaction would not proceed in acetonitrile alone [2]. This would indicate that the role of water is beneficial for promoting the reaction but also detrimental to the hydrolytic instability of phenyl-1,2,4-triazoline-3,5-diones.

Following these results, conjugations were not attempted in completely aqueous conditions; instead, a semi-aqueous system was utilised to improve the solubility and subsequent availability of the tyrosine precursor for reaction whilst effectively minimising the concentration of water (Figure 3). Furthermore, three equivalents of PTAD solution were used and added dropwise rather than all at once, as recommended by Ban et al. [1]. Despite these modifications, no improvement in yield was observed (<5%), suggesting that the electron-withdrawing effect of the –SF_5_ substituent of **5a** was the primary limiting factor influencing the success of the bio-conjugation reaction.

In order to confirm that the hydrolytic instability of **5a** was indeed attributed to the electron-withdrawing effect of the –SF_5_ group, a variety of other *p*-substituted PTAD derivatives **5d** and **5f** were synthesised, added to a solution of 10% acetonitrile/phosphate-buffered saline (pH = 7.4) and analysed periodically over 15-minute intervals by HPLC-MS. To our surprise, the unsubstituted phenyl and even the *p*-methoxy derivative (**5f**) had hydrolysed completely into their corresponding anilines before the first reading (t = 0) was registered by the MS detector due to the relative concentration of water in the dried solvent. However, it should be noted that when these conditions were repeated in the presence of tyrosine, the product was observed within these 15 min, with the p-methoxy derivative (**5f**) affording the highest conversion of the three derivatives tested.

The relatively high yields reported in the literature for the reaction of electron-rich PTAD derivatives with tyrosine in aqueous systems and the rapid hydrolysis rate observed for the *p*-methoxy derivative (**5f**) in phosphate buffer would indicate that the success of the reaction is predicated on the PTAD derivative possessing a greater rate constant for the ene reaction with tyrosine than the competing hydrolysis pathway. [1,2,5] As aromatic substituents become more electron-withdrawing, the hydrolysis pathway evidently begins to predominate. The poor yields obtained with **5a** indicated the requirement for further optimisation of the bio-conjugation reaction. In an effort to find a substitute for the catalytic effect of water, we investigated the PTAD-tyrosine ene reaction across a variety of potentially catalytic lewis acids, protic solvents and bases.

A high-throughput HPLC-MS protocol was developed to determine the most effective catalytic system. 4-[4-(methoxy)phenyl]-1,2,4-Triazole-3,5-dione **(****5f**) was chosen for the study as it was the most stable PTAD derivative, allowing for a less interrupted observation of catalytic activity over the competing hydrolysis pathway.

PTAD derivatives were obtained from the corresponding urazoles using an Oxone/SiO_2_/NaNO_2_ oxidative system (Figure 2). Oxidation of the urazole precursor was performed on a large scale and assumed to be quantitative, and aliquots of known concentration were taken from the oxidation reaction mixture, evaporated under reduced pressure in pre-weighed, flame-dried vials and reconstituted in anhydrous acetonitrile to a concentration of 100 mM, the HPLC-MS auto-sampler was programmed to deliver 10 μL injections of the stock solution into each of the reaction vials containing tyrosine (1 mM, 1 equiv.) and the respective catalyst (2 mM, 2 equiv.). Reactions investigating the effect of alcohol or base were treated with 2 equivalents of the base to ensure both the carboxy terminus and phenol side chain of tyrosine were deprotonated.

Reactions were monitored every 15 min for two hours from the initial injection of the 100 mM PTAD stock solution. Chromatographic peak area of the product Extracted Ion Chromatogram (EIC) peak was determined by manual integration and used as a representation of relative product conversion between individual experiments. The PTAD stock solution could not be analysed via HPLC-MS for degradation between each series of runs, as the PTAD stock hydrolysed immediately in contact with the mobile phase. Therefore, the stock solution was replaced every eight hours to account for any degradation. To determine if any degradation of the PTAD stock solution had occurred over the course of the entire experiment, the stock solution was evaporated to dryness, reconstituted in CDCl_3_ and analysed by ^1^H-NMR, which showed the PTAD stock (**5f**) had not degraded significantly over 72 h (Figure 1).

Acetonitrile was chosen as a solvent for all HPLC-MS test reactions as it minimised unfavourable solvent-Lewis acid and solvent-based interactions among the catalysts investigated compared to alternative polar aprotic solvents. Additionally, Hu et al. reported increased stability of PTAD derivatives in MeCN compared with DMSO and DMF, making it suitable for solubilising the PTAD stock for extended periods for HPLC-MS investigations [5].

The selection of catalysts was guided by prior investigations conducted with azo-dicarboxylate derivatives and the two generally accepted mechanisms for the PTAD-tyrosine reaction, the S_E_Ar-type and Ene-type mechanisms. The wide pH tolerance of the PTAD tyrosine conjugation would indicate that the ene-type pathway is predominating mechanism, as activation of the phenol to the phenoxide under basic conditions does not appear to promote the reaction rate significantly, as one would expect from an S_E_Ar-type pathway (Figure 4). Nevertheless, Solyev et al. identified a number of bases capable of effectively catalysing the reaction between diazocarboxylates and phenolic compounds, achieving yields of up to 99% in 1.5 h, the most successful of which was 5% NaH in THF [9]. When the same NaH/THF system was applied by Al-Momani et al. for the conjugation of [^18^F]4-[4-(sulfonyl fluoride)]-1,2,4-triazole-3,5-dione with both phenol and *N*-acetyl tyrosine methylamide, however, yields of only 28% and 7% were obtained, respectively, indicating the requirement for further investigation towards potential basic catalysts [4]. In addition to NaH, three other bases were chosen to assess the influence of nucleophilicity, basicity and steric hindrance imparted by the subsequent counter-cation of the phenoxide.

It has also been reported that reactions between diazocarboxylates and phenolic compounds are promoted in the presence of protic or Lewis acid catalysts through their interaction with the intermediate anion. Triflic acid was chosen as an organic Bronsted acid, based on prior findings by Leblanc and Boudreault, who used the acid to promote the addition of diazocarboxylate esters to the *para*-position of phenol [10]. Calcium triflate was also chosen as a Lewis acidic salt for comparison. Other Lewis acids investigated included boron trifluoride etherate complex and aluminium trichloride; however, the latter immediately complexed with the tyrosine in solution to form an insoluble precipitate, making it unsuitable for the HPLC-MS catalytic investigation.

Despite extensive efforts to maintain anhydrous reaction conditions for the HPLC-MS protocol prior to injection, the concentration of water, relative to tyrosine and PTAD substrates was substantially higher when compared to that of a batch synthetic or bio-conjugate scale reaction protocol. Consequently, as a function of concentration, the rate constant for hydrolysis is approximately 100-fold greater and substantially out-competes the ene-reaction pathway. Due to the low concentrations of the product obtained, comparisons were made between each catalytic system via chromatographic peak area; rather than translating these values into % mol conversion, exemplary chromatograms can be found in the Appendix A.

Despite this low product conversion, a clear trend is observable among the various catalysts investigated and translatable to a scale suitable for bio-conjugation. It was found that among all the catalysts tested, the highest peak area corresponding to the desired product was obtained 15 min after the initial injection of the PTAD stock solution. After this time point, the peak area slowly depreciated at a relatively consistent rate among all of the catalytic systems tested.

Test reactions were initially performed in aqueous systems for reference; as anticipated, phosphate and Tris-buffered systems and deionized water performed poorly relative to the acetonitrile-based solvent systems. Interestingly, the test reaction containing 2 mM of water in acetonitrile outperformed almost all other test reactions, indicating that water acts at a catalytic capacity at this concentration rather than as a competing nucleophile (Figure 2).

Several alcohols were also investigated as catalytic protic hydrogen bond donors/acceptors of varying nucleophilicity and steric hindrance. Initial predictions were for 2,2,2-trifluoroethanol (TFE) or 1,1,1,3,3,3-hexafluoroisopropanol (HFIP) to be most effective, as they would serve as hydrogen bond donor/acceptors whilst imparting minimal nucleophilic character. The selection of HFIP was further advocated by Tang et al., who found HFIP to be an efficient catalyst for the *para*-substitution of aniline substrates with azodicarboxylates [11].

HFIP in combination with the DIPEA-Tyrosine salt was less successful than initially expected, and increasing the concentration of HFIP two-fold (4 mM) had no significant effect on conversion. However, when the base was omitted from the reaction mixture entirely, a 3-fold increase in chromatographic peak area was observed. A potential explanation for this could be that due to the relatively acidic nature of HFIP as alcohol (pKa = 9.3), the DIPEA cation exchanged between the phenoxide and carboxylate anions and the HFIP, subsequently generating the 1,1,1,3,3,3-hexafluoroisopropoxide. This alkoxide, in turn, may have acted as a nucleophilic species, inducing PTAD’s degradation. This nucleophilic degradation would also explain the higher yields obtained for *t*-butanol rather than *t*-butoxide. However, further investigation is required to confirm this.

All other alcoholic test reactions were performed in the presence of a base (DIPEA, 2 equiv.) as it was suspected the phenol had to be activated irrespective of the hydrogen bonding interactions imparted by the various alcohols; however, the pKa of both methanol and *t*-butanol are both too high to generate substantial quantities of the corresponding alkoxides.

Among the catalytic bases tested, both *t*-BuOK and NaH performed relatively poorly; this could be attributed to the fact that they are insoluble in organic media, forming heterogeneous reaction mixtures. A potential explanation for the success of the caesium carbonate was that minute traces of water in the solvent might have been complexed to the caesium in the form of a hydrate, affording water at a catalytic capacity rather than as a nucleophile moving freely throughout the reaction mixture; CuSO_4_^.^5H_2_O was also chosen to test this hypothesis, but in this case, the reaction performed relatively poorly.

Interestingly, the Lewis acid calcium triflate catalysts outperformed most alcohols and bases, yet BF_3_^.^OEt_2_ was substantially less effective, warranting the investigation of a broader scope of potential Lewis acid catalysts. On the other hand, triflic acid acting in the capacity of a Bronsted acid gave poor results.

Following these results, batch scale reactions in Figure 5 were performed on a 0.25 mmol scale in anhydrous acetonitrile (125 mM) with caesium carbonate (0.5 mmol) under a nitrogen atmosphere (Conditions **A**). Upon addition of the PTAD derivatives, the characteristic red colour dissipated instantaneously in contact with the reaction mixture. Analysis of the crude reaction mixtures revealed very little product conversion and extensive byproduct formation.

The second highest yielding catalytic system (HFIP, without base) was then selected for comparison on a 0.25 mmol scale (Conditions **B**); reactions performed under these conditions were found to be very sluggish, with the characteristic cherry red colour of the PTAD precursors persisting for over 12 h.

In both cases, intractable reaction mixtures were obtained, from which we could not purify the desired product in high purity. Likely, the reaction environment of the HPLC vials and autoinjector system used for the small-scale tests added small aliquots of water that promoted the conjugation reaction; evidently, such conditions were difficult to replicate in a standard round-bottom flask.

While previous investigations of polar aprotic solvents led us to believe acetonitrile would be the most appropriate solvent for facilitating the ene reaction, the sluggish reaction rate observed ruled out its applicability to time-sensitive bio-conjugate protocols, such as those used in radiolabeling.

While DCM is generally not considered compatible with peptides and small polar molecules, its combination with HFIP as a binary mixture has unique solubilising capabilities, which make it a suitable medium for bioconjugate reactions with small peptides and biomolecules without sensitive secondary or tertiary structures. For this reason, the reaction was attempted using a DCM/HFIP solvent system (Conditions **C**). To our delight, it was found that the characteristic cherry red colour dissipated over five minutes for all derivatives. Analysis of the crude products revealed modest-to-high conversion of the tyrosine precursor with minimal byproduct formation.

Attempts to separate the N-acetyl-L-tyrosine precursor from the respective conjugate products proved challenging by means of normal and reverse phase chromatography or recrystallisation. To provide the most accurate representation of each PTAD derivative performance, each product conversion was extrapolated from the crude reaction mixtures by ^1^H-NMR, using the relative integration of the singlet peak corresponding to the un-neighboured aromatic proton, meta to the phenolic hydroxyl group, an example spectrum can be found in the Appendix A. ^1^H-NMR of the crude mixtures indicated a modest-to-high molar conversion across all of the PTAD derivatives investigated (Table 1).

The substantially greater reaction rate of the conjugation in DCM/HFIP compared with MeCN/HFIP would indicate that solvent effects play a substantial role in promoting the reaction. The rapid reaction rate observed in DCM in the absence of base would indicate the ene reaction to be the predominating mechanistic pathway, as the alternative S_E_Ar pathway typically favours polar aprotic media, such as acetonitrile, in order to solubilise the pathways involving charged intermediates, in addition to the base for the activation of the phenol and subsequent substitution of the ortho position (Figure 4). In this solvent system, no clear relation was observable between molar conversion and electronic properties of the various PTAD derivatives. However, as hydrolysis has been eliminated as an influencing factor, the results presented are more likely to be influenced by mechanistic parameters of the ene reaction.

## 3. Conclusions

From these investigations of a variety of catalytic systems, we have found HFIP to be an effective catalyst and water substitute for facilitating the PTAD-tyrosine ene reaction through its action as a non-nucleophilic hydrogen bond donor/acceptor. Interestingly, the HFIP catalyst only appears to be effective when used in tandem with DCM as a solvent, whereas analogous reactions performed in acetonitrile were unsuccessful. This would indicate that future investigations relating to solvent effects are required. The development of this system has allowed for the conjugation of electron-deficient PTAD derivatives to tyrosine in modest-to-high yields, which would otherwise be unachievable under aqueous reaction conditions. Including these electron-deficient PTAD derivatives for use in the PTAD-tyrosine conjugation will hopefully broaden their applicability within fields such as ^19^F-MRI and PET imaging.

## 4. Experimental

General Experimental: All chemical reagents and AR grade or analytical grade solvents were acquired from commercial sources, such as Sigma-Aldrich Merck and Fluorochem. All reactions were monitored using TLC Silica gel 60 F_254_ with UV detection at 254 nm and stained with either iodine. Solvents were removed under reduced pressure using a Buchi Rotavapor rotary evaporator. High-resolution mass spectra were obtained using an Agilent 6510 Q-TOF Mass Spectrometer (ESI). ^1^H-NMR, ^13^CNMR and ^19^F-NMR spectra were recorded on a Bruker Ascend 400 MHz spectrometere (400 MHz ^1^H, 100.6 MHz ^13^C, 376.5 MHz ^19^F). Spectral data for compounds **3a**–**f**, **4a**–**f** and **5a**–**f** can be found in the Appendix A.

General procedure for the synthesis of (**5a**–**f**): A suspension of 4-[4-(pentafluoro-λ⁶-sulfanyl)phenyl]-1,2,4-triazolidine-3,5-dione (0.606 g, 2 mmol), [KHSO_4_-2KHSO_5_-K_2_SO_4_ (1.842 g, 3 mmol)], wet SiO_2_ (50% *w*/*w*) (0.8 g), and NaNO_2_ (0.621 g, 9 mmol) in dichloromethane (12 mL) was stirred at room temperature for 1.5 h and then filtered. Anhydrous Na_2_SO_4_ was added to the filtrate. After 10 min, the resulting mixture was filtered. Dichloromethane was removed by rotary evaporation to obtain the solid product. Experimental data can be found in the Appendix A.

General procedure for the synthesis of (**6a**–**f**): In a 10 mL flask under nitrogen atmosphere, N-acetyl Tyrosine (44 mg, 0.2 mmol) was suspended in anhydrous acetonitrile (2 mL). HFIP (0.4 mmol, 45 uL, 2 equiv.) was then added, and the reaction mixture was allowed to stir for 5 min. Phenyl-1,2,4-triazole-3,5-dione (2 mmol, 1 equiv.) was then reconstituted in anhydrous acetonitrile and added dropwise to the reaction mixture. The reaction was allowed to proceed for 10 min, at which point the solvent was removed under reduced pressure to obtain the crude reaction mixture.

(*S*)-2-acetamido-3-(3-(3,5-dioxo-4-(4-(pentafluoro-λ⁶-sulfanyl)phenyl)-1,2,4-triazolidin-1-yl)-4-hydroxyphenyl)propanoic acid (**6a**) was obtained as a yellow oil, Conversion: 54%. **HRMS (EI)** Calcd for C_19_H_18_F_5_N_4_O_6_S [M+H]^+^ 525.0767, [M+H]^+^ Found 528.0767; **^1^H-NMR ((CD_3_)_2_CO):** δ 8.07 (d, J = 11 Hz, 2H), 7.99 (d, J = 11 Hz, 2H), 7.42 (s, 1H), 7.20 (d, J = 10.5 Hz, 1H), 6.98 (d, J = 10.5 Hz, 1H), 4.72–4.64 (m, 1H), 3.16 (dd, J = 6, 14.5 Hz, 1H), 2.96 (dd, J = 6, 14.5 Hz, 1H), 1.92 (s, 3H); **^13^C-NMR (CD_3_)_2_CO):**.173.13, 170.75, 157.06, 152.00, 136.68, 131.83, 130.14, 128.48, 127.55, 126.41, 124.49, 118.27, 54.40, 37.19, 22.62; ^19^F-NMR (CD_3_)_2_CO): δ 83.97 (m, J = 150.62 Hz, 1F), 62.54 (d, J = 150.62 Hz, 4F). **IR (KBr):** 3315 (br m), 2928 (m), 1769 (s), 1702 (s), 1649 (m), 1513 (m), 1421 (m), 1244 (s), 1102 (w), 1039 (w), 837 (s), 755 (w), 662 (w) 599 (w), 581 (w) cm^−1^.

(*S*)-2-acetamido-3-(3-(3,5-dioxo-4-(4-(trifluoromethyl)phenyl)-1,2,4-triazolidin-1-yl)-4-hydroxyphenyl)propanoic acid (**6b**) was obtained as a yellow oil, Conversion: 76%. **HRMS (EI)** Calcd for C_20_H_18_F_3_N_4_O_6_ [M+H]^+^ 467.1148, [M+H]^+^ Found 467.1148; **^1^H-NMR (****(CD_3_)_2_CO):** δ 7.96 (d, J = 10.5 Hz, 2H), 7.90 (d, J = 10.5 Hz, 2H), 7.41 (s, 1H), 7.19 (dd, J = 2.65, 10.5 Hz, 1H), 6.97 (d, J = 10.5 Hz, 1H), 4.73–4.65 (m, 1H), 3.16 (dd, J = 6, 14.5 Hz, 1H), 2.96 (dd, J = 6, 14.5 Hz, 1H), 1.92 (s, 3H); **^13^C-NMR (****(CD_3_)_2_CO):** δ 173.06, 170.96, 153.13, 152.01, 151.94, 136.82, 131.77, 130.20, 128.28, 124.64, 118.32, 54.49, 49.76, 37.32, 22.66; ^19^F-NMR ((CD_3_)_2_CO): δ - 63.00 (s, 3F); **IR (KBr):** 3084 (br m), 1768 (w), 1693 (s), 1615 (m), 1515 (m), 1418 (m), 1321 (s), 1224 (w), 1123 (s), 1067 (s), 1017 (w), 844 (w), 758 (w), 735 (w), 664 (w), 595 (w), 534 (w) cm^−1^.

(*S*)-2-acetamido-3-(3-(4-(4-fluorophenyl)-3,5-dioxo-1,2,4-triazolidin-1-yl)-4-hydroxyphenyl)propanoic acid (**6c**) was obtained as a yellow oil. Conversion: 62%. **HRMS (EI)** Calcd for C_19_H_18_FN_4_O_6_ [M+H]^+^ 417.1210, [M+H]^+^ Found 417.1207; **^1^H-NMR (****(CD_3_)_2_CO):** δ 7.67–7.64 (m, 2H), 7.38 (s, 1H), 7.29 (t, J = 10, 20 Hz, 2H), 7.17 (dd, J = 5, 10 Hz, 1H), 6.95 (d, J = 10.5 Hz, 1H), 4.70-4.63 (m, 1H), 3.15 (dd, J = 6, 14.5 Hz, 1H), 2.95 (dd, J = 6, 14.5 Hz, 1H), 1.90 (s, 3H); **^13^C-NMR (****(CD_3_)_2_CO):** δ 173.26, 170.06, 163.82, 161.38, 153.69, 152.65, 151.50, 130.18, 129.09. 129.00, 128.73, 127.74, 125.05, 118.35, 116.60, 116.37, 115.89, 54.32, 37.16, 22.60; ^19^F-NMR ((CD_3_)_2_CO): δ − 115.06 (s, 1F); **IR (KBr):** 3080 (br m), 1691 (s), 1510 (s), 1439 (m), 1155 (w), 1014 (w), 839 (m), 759 (w), 662 (w), 597 (w), 520 (w) cm^−1^.

(*S*)-2-acetamido-3-(3-(3,5-dioxo-4-phenyl-1,2,4-triazolidin-1-yl)-4-hydroxyphenyl)propanoic acid (**6d**) was obtained as a yellow oil, Conversion: 68%. **HRMS (EI)** Calcd for C_19_H_19_N_4_O_6_ [M+H]^+^ 399.1305, [M+H]^+^ Found 399.1312; **^1^H**-**NMR (****(CD_3_)_2_CO)**: δ 7.63–7.42 (m, 5H), 7.39 (s, 1H), 7.16 (dd, J = 5, 10 Hz, 1H), 6.95 (d, J = 10.5 Hz, 1H), 4.70-4.63 (m, 1H), 3.15 (dd, J = 6, 14.5 Hz, 1H), 2.95 (dd, J = 6, 14.5 Hz, 1H), 1.90 (s, 3H); **^13^C**-**NMR (****(CD_3_)_2_CO)**: δ 173.05, 170.63, 153.77, 152.73, 151.38, 133.09, 130.23, 128.78, 127.41, 126.91, 125.25, 118.14, 115.94, 54,43, 37.17, 22.68; **IR (KBr):** 3312 (br w), 1695 (s), 1516 (m), 1435 (m), 1233 (s), 823 (s), 769 (s), 697 (s) cm^−1^.

(*S*)-2-acetamido-3-(3-(3,5-dioxo-4-p-tolyl-1,2,4-triazolidin-1-yl)-4-hydroxyphenyl)propanoic acid (**6e**) was obtained as a yellow oil, Conversion: 59%. **HRMS (EI)** Calcd for C_20_H_21_N_4_O_6_ [M+H]^+^ 413.1461, [M+H]^+^ Found 413.1461; **^1^H**-**NMR**
**((CD_3_)_2_CO)**: δ 7.46 (d, J = 10.5 Hz, 2H), 7.37 (s, 1H), 7.31 (d, J = 10.5 Hz, 2H), 7.15 (dd, J = 3, 10.5 Hz, 1H), 6.94 (d, J = 10.5 Hz, 1H), 4.70–4.64 (m, 1H), 3.15 (dd, J = 6, 14.5 Hz, 1H), 2.94 (dd, J = 6, 14.5 Hz, 1H), 2.38 (s, 3H), 1.90 (s, 3H); **^13^C**-**NMR (****(CD_3_)_2_CO)**: δ 173.23, 170.35, 157.04, 153.78, 128.72, 131.14, 130.07, 128.77, 126.84, 118.57, 115.92, 54.47, 37.23, 22.69; **IR (KBr):** 3077 (br, w), 1690 (s), 1514 (s), 1435 (m), 1374 (m), 1285 (w), 1216 (m), 1133 (w), 1102 (w), 815 (w), 760 (w) cm^−1^.

(*S*)-2-acetamido-3-(4-hydroxy-3-(4-(4-methoxyphenyl)-3,5-dioxo-1,2,4-triazolidin-1-yl)phenyl)propanoic acid (**6f**) was obtained as a yellow oil**,** Conversion: 56%. **HRMS (EI)** Calcd for C_20_H_21_N_4_O_7_ [M+H]^+^ 429.1410, [M+H]^+^ Found 429.1408; **^1^H**-**NMR**
**((CD_3_)_2_CO)**: δ 7.48 (d, J = 10.5 Hz, 2H), 7.38 (s, 1H), 7.15 (dd, J = 3, 10.5 Hz, 1H), 7.07–7.03 (d, 2H), 6.94 (d, J = 10.5 Hz, 1H), 4.71–4.62 (m, 1H), 3.15 (dd, J = 6, 14.5 Hz, 1H), 2.94 (dd, J = 6, 14.5 Hz, 1H), 1.90 (s, 3H); **^13^C**-**NMR (****(CD_3_)_2_CO)**: δ 173.26, 170.38, 170.19, 160.26, 157.07, 150.89, 131.12, 130.24, 128.77, 125.53, 118.66, 115.94, 114.92, 55.84, 54.57, 37.35, 22.64; **IR (KBr):** 3080 (br, w), 1691 (s), 1812 (w), 1513 (s), 1443 (m), 1368 (w), 1301 (w), 1248 (m), 1177 (m), 1021 (w), 892 (w), 760 (w) cm^−1^.

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
