# Peer review of "Novel Strategy for Non-Aqueous Bioconjugation of Substituted Phenyl-1,2,4-triazole-3,5-dione Analogues"

_molecules, 2022, doi:10.3390/molecules27196667_

Round 1

Reviewer 1 Report

The authors have synthesized a range of Phenyl-1,2,4-triazoline-3,5-diones (PTADs) derivatives with para substituents of varying electron donating and withdrawing groups to investigate correlation between substituent electronic effects and relative hydrolytic stabilities. The bio-conjugation of those derivatives with tyrosine amino acid were performed in both the aqueous and organic media in the presence of different catalysts. The work seems informative in the field of bio-conjugation. However, the manuscript requires major revision addressing the following comments:

1.     The abstract should contain more information related to the results. Currently it seems like introduction.

2.     The work seems lacking novelty. The same reaction was already reported in different pH. The problem-statement of the work has also been discussed in other reports. This work seems a continuation of the earlier work. The author should give justification on the novelty in the manuscript.

3.     Page 7, Line 222: ‘exemplary chromatograms can be found in the Supplementary Materials section’. No figure numbers are mentioned for this statement. Moreover no chromatogram is shown in the supporting information.

4.     The author should discuss how the peak area was calculated? The full chromatogram of all the products should be given in supporting information.

5.     Scheme 1 shows the applicability of the work in protein. However, the authors presented all the work with tyrosine molecule. It is important to see how this work applies in protein as that’s the ultimate goal. The authors should show atleast one reaction in protein or peptide containing tyrosine.

6.     In Figure 2: Were the MeOH, t-BtOH, DIPEA, NaH used as catalysts?

7.     Page 3, line 83: (Scheme 2) should be before the full stop.

Author Response

We thank the reviewers for the useful comments and suggestions. The modifications and point-by-point responses are reported below.

Reviewer 1:

The authors have synthesized a range of Phenyl-1,2,4-triazoline-3,5-diones (PTADs) derivatives with para substituents of varying electron donating and withdrawing groups to investigate correlation between substituent electronic effects and relative hydrolytic stabilities. The bio-conjugation of those derivatives with tyrosine amino acid were performed in both the aqueous and organic media in the presence of different catalysts. The work seems informative in the field of bio-conjugation. However, the manuscript requires major revision addressing the following comments:

  1. The abstract should contain more information related to the results. Currently it seems like introduction.

Reply 1: We understand the point and we agree with the observation. Therefore, the abstract content has been updated to better reflect the results of the paper.

  1. The work seems lacking novelty. The same reaction was already reported in different pH. The problem-statement of the work has also been discussed in other reports. This work seems a continuation of the earlier work. The author should give justification on the novelty in the manuscript.

Reply 2: While this type of conjugation has been utilised previously, its applicability has been limited to electron-rich aromatic PTAD derivatives and the issue of hydrolytic instability has not been investigated thoroughly. We believe that the novelty of this work arises from a more methodical study of the issue and from the identification of a solvent system allowing successful conjugation of electron-deficient derivatives; this result would not be achievable under the reaction conditions previously described in the literature.

  1. Page 7, Line 222: ‘exemplary chromatograms can be found in the Supplementary Materials section’. No figure numbers are mentioned for this statement. Moreover no chromatogram is shown in the supporting information.

Reply 3: We thank the reviewer for highlighting our miss. Exemplary chromatograms have now been included in supplementary materials.

  1. The author should discuss how the peak area was calculated? The full chromatogram of all the products should be given in supporting information.

Reply 4: The following statement has been added: “Chromatographic peak area of the product Extracted Ion Chromatogram (EIC) peak was determined by manual integration and used as a representation of relative product conversion between individual experiments.”. Examples of Extracted and Total Ion Chromatograms have been included in the supplementary materials section.

  1. Scheme 1 shows the applicability of the work in protein. However, the authors presented all the work with tyrosine molecule. It is important to see how this work applies in protein as that’s the ultimate goal. The authors should show at least one reaction in protein or peptide containing tyrosine.

Reply 5: While the conjugation of these unstable compounds to peptides and protein is the intended final application of this project, the primary objective of this paper is to identify and address the limitations of the PTAD conjugation using model systems, to provide researchers with the optimal parameters for tagging biomolecules with these compounds in future works.

  1. In Figure 2: Were the MeOH, t-BtOH, DIPEA, NaH used as catalysts?
  2. Page 3, line 83: (Scheme 2) should be before the full stop.

Reply 6 and 7: These issues have been corrected in the manuscript.

Reviewer 2 Report

A series of phenyl-1,2,4-triazoline-3,5-diones (PTADs) were synthesized and their reactivity in aqueous and organic solvent systems (MeCN and DCM) were compared. The work was solidly performed. I especially appreciate the presentation of many negative results, which is unusual, yet extremely helpful for others to follow the work.

I will suggest the author to include a table in the manuscript to list all reaction conditions being tested, and give the results.

Also, I don’t understand how the authors could draw the conclusion that: “Including these electron-deficient PTAD derivatives for use in the PTAD-Tyrosine conjugation will hopefully broaden their applicability within fields such as 19F-MRI and PET imaging.” – Considering the reaction cannot be performed in aqueous media, how such molecules could be used? Clarification is necessary.

Author Response

We thank the reviewers for the useful comments and suggestions. The modifications and point-by-point responses are reported below.

Reviewer 2:

A series of phenyl-1,2,4-triazoline-3,5-diones (PTADs) were synthesized and their reactivity in aqueous and organic solvent systems (MeCN and DCM) were compared. The work was solidly performed. I especially appreciate the presentation of many negative results, which is unusual, yet extremely helpful for others to follow the work.

  1. I will suggest the author to include a table in the manuscript to list all reaction conditions being tested, and give the results.

Reply 8: While we understand the general utility of having a synoptic table of the conditions employed, in our case conditions A and B provided little to no yield, and we think that reporting the results in-text would be sufficient. However, we take the suggestion of the reviewer and add such table in the Supp Info section, as reported below.

The three conditions were tested on a synthetic scale based on results obtained from LC-MS catalytic experiments. Condition A: 5a decomposed rapidly to give numerous by-products and < 5% yield of 6a, Condition B: Reaction was very slow (> 12 hrs), rendering it unsuitable for the purposes of bioconjugation, Condition C: Reaction proceeded rapidly with minimal by-products formation.

Reaction (6a)

Conditions

Yield (%)

A

MeCN, Cs2CO3 (2 equiv.)

< 5

B

MeCN, HFIP (2 equiv.)

No reaction

C

DCM, HFIP (2 equiv.) 

54

  1. Also, I don’t understand how the authors could draw the conclusion that: “Including these electron-deficient PTAD derivatives for use in the PTAD-Tyrosine conjugation will hopefully broaden their applicability within fields such as 19F-MRI and PET imaging.” – Considering the reaction cannot be performed in aqueous media, how such molecules could be used? Clarification is necessary.

Reply 9: We think to understand the confusion highlighted by the reviewer and would like to clarify. While the conjugation must be performed in non-aqueous media, the final product will require purification prior to administration and use in imaging; this is a standard practice for the use PET tracers, and the requirement for purification would not reduce the value of this bio-conjugative approach.

Round 2

Reviewer 1 Report

The authors addressed all my comments. I think the manuscript can be published as its current form.